# *NRXN1* as a Prognostic Biomarker: Linking Copy Number Variation to EMT and Survival in Colon Cancer

**DOI:** 10.3390/ijms252111423

**Published:** 2024-10-24

**Authors:** Hyun Jin Bang, Hyun-Jeong Shim, Mi-Ra Park, Sumin Yoon, Kyung Hyun Yoo, Young-Kook Kim, Hyunju Lee, Jeong-Seok Nam, Jun-Eul Hwang, Woo-Kyun Bae, Ik-Joo Chung, Eun-Gene Sun, Sang-Hee Cho

**Affiliations:** 1Division of Hematology and Oncology, Department of Internal Medicine, Chonnam National University Medical School and Hwasun Hospital, Hwasun 58128, Republic of Korea; daunjin@gmail.com (H.J.B.); hjnhj@chonnam.ac.kr (H.-J.S.); fang79@naver.com (M.-R.P.); juneul2@gmail.com (J.-E.H.); drwookyun@chonnam.ac.kr (W.-K.B.); ijchung@chonnam.ac.kr (I.-J.C.); 2Laboratory of Biomedical Genomics, Department of Biological Sciences, Sookmyung Women’s University, Seoul 04310, Republic of Korea; a435103@sookmyung.ac.kr (S.Y.); stellajjun@gmail.com (K.H.Y.); 3Department of Biochemistry, Chonnam National University Medical School, Hwasun 58128, Republic of Korea; ykk@chonnam.ac.kr; 4School of Electrical Engineering and Computer Science, Gwangju Institute of Science and Technology, Gwangju 61005, Republic of Korea; hyunjulee@gist.ac.kr; 5School of Life Sciences, Gwangju Institute of Science and Technology, Gwangju 61005, Republic of Korea; namje@gist.ac.kr; 6National Immunotherapy Innovation Center, Chonnam National University Medical School, Hwasun 58128, Republic of Korea

**Keywords:** colorectal cancer (CRC), mechanisms of inhibition, epithelial–mesenchymal transition

## Abstract

The role of biomarkers in cancer treatment varies significantly depending on the cancer stage. Thus, in clinical practice, tailoring biomarkers to meet the specific needs and challenges of each cancer stage can increase the precision of treatment. Because they reflect underlying genetic alterations that influence cancer progression, copy number variation (CNV) biomarkers can play crucial prognostic roles. In our previous study, we identified potential survival-related genes for colorectal cancer (CRC) by analyzing CNV and gene expression data using a machine-learning approach. To further investigate the biological function of NRXN1, we assessed the use of RNA sequencing, phosphokinase assays, real-time quantitative PCR, and Western blot analysis. We found that *NRXN1* copy number deletion was significantly associated with poor overall survival (OS) and recurrence-free survival (RFS), even in patients who received adjuvant chemotherapy. Compared with its expression in normal tissues, *NRXN1* expression was lower in tumors, suggesting its potential role as a tumor suppressor. *NRXN1* knockdown enhanced CRC cell viability and invasion, and transcriptome analysis indicated that the increased invasion was caused by GSK3β-mediated epithelial–mesenchymal transition. These findings highlight *NRXN1* copy number deletion as a novel biomarker for predicting recurrence and survival in patients with resected colon cancer.

## 1. Introduction

Colorectal cancer (CRC) is the second leading cause of cancer-related deaths and the fourth most common cancer in the United States [1]. Currently, after surgery, adjuvant chemotherapy is the standard treatment for preventing recurrence in patients with stage III and high-risk stage II CRC [2]. However, recurrence is still a major concern, and progress in the identification of new prognostic indicators for truly high-risk patients, as well as advancements in adjuvant chemotherapy, has been limited. Recently, examining circulating tumor DNA (ctDNA) after surgical resection has shown promise for identifying patients at risk of recurrence and, potentially, for predicting the benefits of adjuvant chemotherapy [3]. However, although ctDNA is a valuable biomarker, predictive recurrence biomarkers are lacking. Therefore, to predict recurrence, a comprehensive understanding of genomic and molecular characteristics is crucial. Because colon cancer chemotherapeutic options are limited after surgery, novel therapies that more effectively reduce recurrence and improve survival are essential [4]. Therefore, the identification of novel recurrence-associated genes in clinical samples and the elucidation of their mechanisms of action could meet these unmet needs.

Cancer develops through various mechanisms that involve the tumor microenvironment, angiogenesis, and genetic changes, including genetic variations, single nucleotide polymorphisms, and chromosome aberrations [5,6]. Large high-throughput cancer sequencing datasets, including genomic, proteomic, transcriptomic, and epigenomic data, are available for research. Machine learning (ML) is a crucial tool for comprehensively analyzing such vast datasets and exploring potential biomarkers [7].

Single nucleotide polymorphisms, the predominant type of genomic variation, are known to cause phenotypic variations that are responsible for tumorigenesis [8]. However, recent studies have shown that by altering gene expression levels, oncogene or tumor suppressor gene (TSG) copy number variation (CNV) is strongly associated with cancer development and progression [9]. Additionally, because of their high stability, gene CNVs are considered robust cancer biomarkers.

Previously, by analyzing CNV and gene expression data via ML, we identified 14 putative survival-associated genes in patients with colon cancer after surgery. Screening for pathogenic survival-associated driver genes and validation via in silico system analysis on the basis of clinical genomic data identified Rab GTPase-activating protein 1-like (*RABGAP1L*), myosin heavy chain 9 (*MYH9*), and dopamine receptor D4 (*DRD4*) as recurrence-associated candidate genes with protumoral potential in vitro. Finally, a gene expression-based recurrence prediction model was developed [10]. Our data suggest a new gene to predict survival and identify previously unknown tumorigenesis-associated genes. However, the prognostic value of ML-identified biomarker genes requires experimental validation, and the elucidation of their precise mechanisms of action to uncover new druggable targets or for drug recreation is a very promising approach to improving treatment outcomes.

In this direction, we designed this study to identify survival-associated CNV biomarkers in patients who had undergone colon cancer surgery using an ML platform and to validate their potential as biomarkers by analyzing their functional roles in tumorigenesis.

## 2. Results

### 2.1. Identification of Putative CRC Prognostic Markers

Previously, using an ML platform and openly accessible data, including CNV and gene expression data, we identified 14 putative survival-associated genes in patients with CRC (Figure 1A) [10]. To determine their prognostic value, we conducted experimental CNV and gene expression validation analyses using matched colon cancer samples. The CNV analysis involved 137 patients, 75 (55%) and 62 (45%) of whom had stage II and III CRC, respectively. Two patients died with stage II, and 10 died with stage III. Due to the limited number of events observed for survival analysis in the stage II group, survival analysis was conducted only in the stage III group. Survival analysis based on CNV analysis revealed that *NRXN1* and *WDR72* copy number deletion (CND) were significantly associated with poor RFS and OS, whereas *KIF1B* CND was significantly associated with poor RFS only (Figure 1B). Among these genes, *NRXN1* CND showed the highest significance. Next, gene expression analysis of matched patient samples was used to determine the functional role of these genes. This analysis revealed that *NRXN1* and *WDR72* expression levels were lower in tumors than in normal tissues, whereas *KIF1B* gene expression was higher in tumors (Figure 1C). The extended gene expression analysis data are shown in Appendix A. These results suggest that *NRXN1* CND is a potential prognostic marker after colon cancer resection and indicate that *NRXN1* might be a TSG. Therefore, further analyses were performed to elucidate the function of *NRXN1* in colon cancer tumorigenesis.

### 2.2. Correlation Between NRXN1 CNV and Differential Gene Expression

Somatic CNVs may influence cancer development and progression by altering gene expression. Therefore, we investigated the relationship between CNV and *NRXN1* gene expression in resected stage II and III colon cancer samples. This analysis revealed that *NRXN1* gene expression tended to correlate with the CNV status of *NRXN1*, such as its amplification, deletion, and absence (Figure 2A). Colon cancer samples with *NRXN1* CND presented lower gene expression than those without CNV or copy number amplification.

### 2.3. Correlation Between NRXN1 CND and Response to Adjuvant Chemotherapy

Next, we analyzed the associations between *NRXN1* CND and survival outcomes in patients with CRC. Patients with *NRXN1* CND had worse RFS and OS figures than those without the deletion, whether or not they received adjuvant chemotherapy, such as 5-fluorouracil and oxaliplatin (Figure 2B). These results suggest that *NRXN1* CND is linked to tumor recurrence and poor survival outcomes. Even within the adjuvant chemotherapy group, a patient with *NRXN1* CND has significantly worse survival, implying that *NRXN1* CND may contribute to chemotherapy resistance. Despite receiving standard chemotherapy, the survival benefit for patients with *NRXN1* CND was minimal, indicating that *NRXN1* CND might hinder chemotherapy efficacy. Therefore, *NRXN1* CND could serve as a poor prognostic biomarker in CRC.

### 2.4. NRXN1 Knockdown Enhances CRC Cell Viability and Significantly Increases Invasion

Next, we knocked down *NRXN1* in CRC cell lines using siRNA to reproduce CND and investigate the functional significance of *NRXN1*. We confirmed the successful knockdown of NRXN1 in the siNRXN1-transfected cells compared to control cells using RT-qPCR and Western blot analysis, as shown in Appendix A. Compared to siNC-transfected control cells, siNRXN1-transfected CRC cell lines exhibited a slight increase in viability (Figure 3A). However, analysis of the effect of NRXN1 knockdown on cancer cell invasion revealed a significantly increased invasive capacity in CRC cell lines, such as HCT116, NT29, and Caco2, upon *NRXN1* knockdown (Figure 3B). These results indicate that while *NRXN1* knockdown slightly promotes CRC cell viability, it has a more pronounced effect on enhancing cell invasion.

### 2.5. Transcriptome Analysis of NRXN1-Silenced CRC Cells

Next, to assess the mechanism underlying the effects of *NRXN1* knockdown in CRC cells, RNA-seq analysis was performed in siNC- or siNRXN1-transfected HCT116 cells. This analysis revealed that 320 genes were upregulated and 623 were downregulated in siNRXN1-transfected cells compared with siNC-transfected controls (fold change > 2, *p* < 0.05; Figure 4A). Reactome pathway analysis revealed that the upregulated genes were associated with the response to growth factors, epithelial cell proliferation, Wnt signaling, and mesenchymal cell proliferation, whereas the downregulated genes were associated with cell junction assembly, epithelial cell proliferation and differentiation, and cell–cell junction organization (Figure 4B). Consistent with the changes in invasion, *NRXN1* knockdown modulated genes associated with epithelial/mesenchymal cell proliferation and cell junctional organization, indicating that *NRXN1* knockdown affects epithelial–mesenchymal transition (EMT) (Figure 4C). ZEB1 induces EMT during cancer metastasis by transcriptionally repressing CDH1, which encodes E-cadherin [11]. Transcriptome analysis revealed that *NRXN1* knockdown induced ZEB1 expression, which was accompanied by the downregulation of CDH1 and epithelial cell adhesion molecules. These data show that *NRXN1* knockdown in CRC cells promotes EMT and invasion.

### 2.6. NRXN1 Knockdown Induces EMT in CRC Cells

To validate the transcriptome analysis findings, we investigated the effect of *NRXN1* knockdown on the EMT signaling pathway in CRC cells, and observed increased ZEB1, N-cadherin, snail, slug, and twist expression in siNRXN1-transfected CRC cells, whereas E-cadherin levels decreased at the mRNA and protein levels (Figure 5A,B). Consistent with the findings from transcriptome analysis and phenotypic changes in invasion properties, Western blot and RT-qPCR analyses revealed that *NRXN1* knockdown promotes EMT in CRC cells.

### 2.7. NRXN1 Knockdown Induces GSK3β Phosphorylation

Next, we used human phosphokinase array analysis of lysates from siNC- or siNRXN1-transfected CRC cells to identify the signaling pathway induced by *NRXN1* knockdown. This analysis revealed increased GSK3β (S9) and p53 (S46) phosphorylation, as well as decreased HSP60 and p53 (S15) phosphorylation, in siNRXN1-transfected HCT116 cells. HT29 CRC cells exhibited increased GSK3β (S9) phosphorylation and decreased Chk-2 (T68) and p53 (S15) phosphorylation upon NRXN1 knockdown. However, no change was detected in the phosphorylation of STAT3 (Y705, S727), a major EMT regulator (Figure 6A). NRXN1 knockdown-mediated GSK3β phosphorylation was also validated via Western blot analysis. Collectively, these findings indicate that NRXN1 knockdown in HCT116, HT29, and Caco2 cells induces GSK3β (S9) phosphorylation.

### 2.8. GSK3β Mediates EMT and Invasion upon NRXN1 Knockdown in CRC Cells

Next, we assessed whether GSK3β activation mediates EMT and invasion, and whether GSK3β inhibitors could be potential treatment candidates for *NRXN1*-knockdown CRC cells. Our analysis revealed that treatment with the GSK3β inhibitor SB216763 effectively inhibited the upregulation of ZEB1, N-cadherin, slug, and snail (Figure 7A). GSK3β inhibition also restored E-cadherin expression in *NRXN1*-knockdown CRC cells. Furthermore, GSK3β inhibition modulates the invasive phenotypic changes induced by NRXN1 knockdown (Figure 7B), effectively blocking the enhanced invasion of CRC cells. Moreover, consistent with the effects of the GSK3β inhibitor SB216763, siRNA-mediated knockdown of GSK3β effectively suppressed the invasion properties and EMT induced by NRXN1 knockdown (Appendix A). These findings indicate that *NRXN1* knockdown induces EMT and invasion via GSK3β activation and identifies GSK3β inhibitors as potential therapeutic candidates for *NRXN1* downregulation in colon cancer.

## 3. Discussion

Neurexins, a family of highly polymorphic presynaptic cell adhesion molecules, are involved in synaptic transmission. Genomic alterations in *NRXN* genes (*NRXN1, NRXN2, and NRXN3*) are associated with neuropsychiatric disorders, including autism spectrum disorders and schizophrenia [12,13,14]. *NRXN1* has been implicated in various neuropsychiatric and neurodevelopmental diseases [15,16]. Moreover, Yotsumoto et al. identified *NRXN1* as a tumor-specific marker and reported that when it is overexpressed on the cell surface, it is a novel molecular target for the development of effective antibody drug conjugates (ADCs) against small-cell lung cancer [17]. Several studies have associated *NRNX1* with poor outcomes, disease progression, and distant metastasis, proposing it as a potential independent predictor of Ewing sarcoma and breast cancer [18,19]. To our knowledge, this is the first study to report a role for *NRXN1* in colon cancer tumorigenesis. Using an ML approach, we found that *NRXN1* CND is associated with poor clinical prognosis after colon cancer resection. Additionally, we showed that *NRXN1* CND promotes EMT through GSK3β activation.

For cancer biomarker development, CNVs have several advantages over other types of gene expression data [20]. Crucially, CNVs are more stable than gene expression levels, which can fluctuate because of various factors, including environmental factors, the cell cycle stage, and technical sample-processing variations. Therefore, they are more reliable and reproducible cancer prognostic and diagnostic biomarkers. Additionally, by increasing or decreasing gene copy numbers, CNVs directly alter gene dosage, which can significantly and immediately impact cell function and cancer progression. Therefore, understanding CNVs can help predict cancer aggressiveness, spreading potential, and treatment response, which can guide the development of new targeted therapies that specifically address these genetic alterations [21,22].

EMT, a biological reprogramming process that confers a mesenchymal phenotype to epithelial cells, is crucial for embryogenesis, wound healing, and cancer progression [23,24,25]. EMT programs induce tight junction dissolution, apical–basal polarity disruption, and cytoskeletal architecture reorganization, which can promote cancer cell migration, invasion, and extensive heterogeneity [26,27]. EMT activation is regulated by the tumor microenvironment, including growth factors, inflammatory cytokines, and transcription factors, which lead to increased therapy resistance [28,29]. EMT plays key roles in colon cancer progression and the prevention of colon cancer invasion, recurrence, and metastasis [30,31]. Furthermore, in patients with high-risk stage II or III CRC, traditional clinicopathological features or molecular signatures alone are insufficient for predicting RFS across different subgroups. Using public datasets, recent reports have shown that EMT-associated gene signatures can predict RFS in patients with stage II/III CRC [32,33]. Therefore, to guide clinical decision making, there is a need for more accurate recurrence-related biomarkers and risk stratification optimization.

This study identified *NRXN1* as a new marker that regulates the EMT-associated GSK3β pathway during tumorigenesis. High GSK3β expression is reported to be correlated with worse breast cancer and CRC outcomes [34,35]. Additionally, GSK3β plays an essential role in glioblastoma carcinogenesis and is involved in brain function through the modulation of various processes, such as neuronal morphology, synapse formation, neuroinflammation, and neurological disorders [36]. In particular, GSK3β phosphorylation at Ser9 is known to play a pivotal role in regulating EMT. When GSK3β is phosphorylated, its suppression of transcription factors like snail and slug is relieved, leading to the promotion of EMT. These mechanisms contribute to cancer cell invasion and metastasis [34,37,38]. In our study, NRXN1 knockdown was shown to induce GSK3β phosphorylation at Ser9, which aligns with the established role of GSK3β in cancer progression. Therefore, further studies are needed to determine the role of *NRXN1* and GSK3β in extraneuronal tumorigenesis.

A key strength of our study is that *NRXN1* CND is associated with poor survival outcomes, regardless of chemotherapy, indicating a potential link to chemotherapy resistance. Additionally, EMT, which is known to play a role in tumorigenesis, has also been implicated in chemotherapy resistance [39,40,41]. Therefore, it is necessary to focus on the role of *NRXN1* CND in colon cancer EMT and the context of chemotherapy resistance.

Although their therapeutic targeting has limitations, TSGs significantly impact cancer progression because they are involved in angiogenesis, signal transduction, and the development of chemotherapy resistance [42]. To further investigate the roles of TSGs, an in-depth understanding of the molecular mechanisms involved is needed. In addition to the simple two-hit model, TSG inactivation involves epigenetic regulation via phosphorylation and transcriptome regulation [43]. Because targeting TSGs is more complex than targeting oncogenes, an alternative approach is to target the downstream signaling pathways that are altered by TSG inactivation [44].

Our findings show that *NRXN1* functions as a TSG in colon cancer and suggest that GSK3β inhibition is a promising therapeutic option in the context of *NRXN1* knockdown; however, further validation studies are needed. Because CNVs can also be analyzed in ctDNA from blood samples, validating these findings through noninvasive liquid biopsies can provide valuable insights into risk prediction and stratification.

The retrospective design and small sample size of stage III colon cancer patients in this study may limit the generalizability and robustness of our findings. To validate our results and gain a better understanding of the potential role of NRXN1 in CRC, prospective large-scale studies are needed to validate and understand the potential role of *NRXN1* in CRC.

## 4. Materials and Methods

### 4.1. Data Processing and ML Algorithms

The details of the ML algorithm analytic methods were described in our previous study [10]. The in-house test dataset for the prediction model included 137 patients with stage II and III colon cancer who had undergone curative resection between January 2013 and December 2014.

### 4.2. Biospecimens and Genomic DNA Extraction

Among the samples from the 137 patients, the expression of putative survival-associated genes was evaluated in 120 stage II and III colon cancer samples with matched tumor and normal tissue pairs. This study’s ethical approval was granted by Chonnam National University Hwasun Hospital’s institutional review board (approval number CNUHH-2020-173). The biospecimens were obtained with informed consent from Chonnam National University Hwasun Hospital’s biobank, which is a member of the Korea Biobank Network. For CNV analysis, genomic DNA was extracted using a tissue DNA kit (Cat No. 51404; Qiagen, Valencia, CA, USA) following the manufacturer’s protocol.

### 4.3. Library Preparation for CNV Analysis

Library preparation for an 81-gene panel was performed via a customized QIAseq targeted DNA panel (Qiagen) according to the manufacturer’s instructions. The panel, which was designed to facilitate the amplification-based capture and sequencing of the coding regions of 81 cancer-associated genes, includes a pool of 3332 primers, and 20 ng of template DNA is needed per sample. Sample barcoding was performed using a QIAseq 12-Index L kit (Qiagen).

### 4.4. Templating and Sequencing

Fully automated template preparation was performed on an Ion Chef™ System (Thermo Fisher Scientific, Waltham, MA, USA) after library quantification on a 4150 TapeStation System (Agilent Technologies, Santa Clara, CA, USA). Next, equal library amounts were pooled and loaded onto Ion 540™ chips (Thermo Fisher Scientific) using an Ion Chef™ System (Thermo Fisher Scientific), followed by sequencing on an Ion S5XL™ System (Thermo Fisher Scientific) using an Ion S5™ Sequencing Kit (Thermo Fisher Scientific) according to the manufacturer’s instructions.

### 4.5. Data Processing and Analysis

Sequence read alignment with the reference genome and base calling were performed via Torrent Suite software v5.8.0 (Thermo Fisher Scientific). FASTQ files were generated using the FileExporter Plugin v5.8.0.2 (Thermo Fisher Scientific). After downloading from the Ion Torrent server, raw FASTQ files were imported into the CLC Genomics Workbench program v21 (Qiagen) [45]. Because of the Ion Torrent sequencing data and custom Qiagen panel features, a custom workflow was used for the analysis. The FASTQ files were aligned to the human reference genome HG19 using default settings, and the aligned BAM files with a target amplicon coverage of over 700× were used for downstream analysis.

### 4.6. Somatic Variants and CNV Calling

Paired aligned tumor and normal tissue BAM files were used for somatic variant calling via default settings. The workflow of the CNV detection module in the CLC workbench was used for CNV analysis, with a minimum fold change of 1.4, a significance threshold of 0.05, and a low coverage cutoff of 30 [46]. A gene-level annotation track file (Gene CNVs) containing CNVs with a false discovery rate of <0.01 was used for downstream analysis.

### 4.7. Cell Culture and Reagents

The CRC lines HCT116, HT29, Caco2, HCT15, and Colo205 were purchased from the American Type Culture Collection (Manassas, VA, USA) and the Korean Cell Line Bank (Seoul, Republic of Korea). The cells were cultured in DMEM or RPMI-1640 supplemented with 10% fetal bovine serum, penicillin (100 units/mL), and streptomycin (100 μg/mL) (Invitrogen, Carlsbad, CA, USA) in a humidified incubator at 37 °C and 5% CO_2_. Universal nontargeting siRNA (negative control, NC) and siNRXN1 (Cat. No. SR306214) were purchased from OriGene Technologies, Inc. (Rockville, MD, USA). For GSK3β knockdown, we used GSK3β siRNA (Catalog #2932) purchased from Bioneer Corporation (Daejeon, Republic of Korea). For siRNA transfection, lipofectamine RNAiMAX (Cat. No. 13778150; Invitrogen, Carlsbad, CA, USA) was used following the manufacturer’s protocol. SB216763 (Cat. No. S3442), a GSK3β inhibitor, was purchased from Sigma–Aldrich (St. Louis, MO, USA).

### 4.8. Cell Viability Assay

A cell counting kit-8 assay (Cat. No. 96992, Sigma–Aldrich) was used for cell viability analysis according to the manufacturer’s instructions. Briefly, cells were seeded in 96-well plates and cultured for 16–24 h. They were then transfected with siRNA for 48 h, followed by the addition of the cell counting kit-8 reagent (10%, *v*/*v*) into each well and incubation for 1 h at 37 °C with 5% CO_2_. Finally, the optical density was measured at 450 nm via a SpectraMax i3X microplate reader (Molecular Devices, Sunnyvale, CA, USA).

### 4.9. RNA Sequencing and Data Analysis

Cutadapt (version 4.1) was used to process reads by removing adapter sequences and low-quality segments. The trimmed reads were then aligned to the human reference genome (version hg19) using the STAR aligner (version 2.6.1c), and read counts were calculated via rsem-calculate-expression (version 1.3.1) with the default settings. Then, we used DESeq2 software (version 1.38.3) to normalize the raw counts and obtain differentially expressed genes (DEGs). DEGs were defined as genes with over 2-fold change (FC), a *p*-value of less than 0.05, an average expression value greater than 5 in at least one group, and an average in each group lower than the standard deviation of each group. Functional annotations and enrichment analyses of DEGs were performed using Metascape, a powerful web-based tool.

### 4.10. Invasion Assay

Cell invasion was assessed using a Transwell assay. Equal numbers (1 × 10^5^) of control and siRNA-transfected cells were seeded in 24-well Transwell filter chambers coated with 1 μg/mL Matrigel (Cat. No. 354234; Corning, Bedford, MA, USA). Cell culture medium containing bovine serum albumin (5%) and 20 μg/mL of the chemoattractant fibronectin (Calbiochem, La Jolla, CA, USA) was added to the bottom chamber. The cells were then incubated for 24 h before being stained with Diff-Quick staining reagent (Cat. No. 38721; Sysmex Corporation, Kobe, Japan). ImageJ (National Institutes of Health, Bethesda, MD, USA) was then used to quantify the invaded cell area.

### 4.11. Real-Time Quantitative Reverse Transcription PCR (RT-qPCR)

RT-qPCR was used for gene expression analysis. Total RNA isolation and cDNA synthesis were performed using Hybrid-R reagent (Cat. No. 305–101; GeneAll Biotechnology, Seoul, Republic of Korea) and a GoScript kit (Cat. No. A5003; Promega, Madison, WI, USA), respectively, following the manufacturers’ protocols. RT-qPCR analysis was performed on a CFX96 real-time PCR detection system (Bio-Rad Laboratories, Hercules, CA, USA) using SYBR Green Supermix and the primers listed in Appendix A. Fold changes in gene expression were calculated via the ΔΔCT method, with GAPDH used as the reference gene.

### 4.12. Western Blot Analysis

The cells were subsequently washed with phosphate-buffered saline. Whole-cell lysates (WCLs) were prepared using M-PER mammalian protein extraction reagent (Cat. No. 78051; Thermo Fisher Scientific) supplemented with a protease and phosphatase inhibitor cocktail (Cat. No. 1861281; Thermo Fisher Scientific). A Pierce BCA protein assay kit (Thermo Fisher Scientific) was used to quantify the WCL protein levels. Equal amounts of WCL were used for sodium dodecyl sulfate–polyacrylamide gel electrophoresis, followed by protein transfer onto polyvinylidene fluoride membranes. The membranes were then blocked with SuperBlockTM T20 blocking buffer (Cat. No. 37515; Thermo Fisher Scientific) and incubated with primary antibodies (Appendix A) at 4 °C overnight. They were then probed with horseradish peroxidase-conjugated secondary antibodies for one hour at 25 °C and visualized using an imaging system (LAS-4000 mini, Fujifilm, Tokyo, Japan). Band intensities were quantified via ImageJ (National Institutes of Health).

### 4.13. Phosphokinase Array

A phosphokinase array was performed on HCT116 and HT29 cells transfected with siRNA-NC or siNRXN1 for 48 h via a Proteome Profiler Human Phospho-Kinase Array Kit (Cat. No. ARY003C, R&D Systems, Minneapolis, MN, USA). Briefly, the kit-provided lysis buffer containing a protease and phosphatase inhibitor was used for total protein extraction, followed by protein quantification. Next, the array membranes were incubated overnight with a mixture of cell lysates (300 µg)/array buffer 1 at 4 °C. After detection of antibody reaction and streptavidin-HRP incubation, the array membranes were visualized using a low-light imaging system (LAS-4000 mini). The signal intensities were quantified via ImageJ software (version 1.53k, National Institutes of Health, Bethesda, MD, USA), and the results were confirmed via Western blotting.

### 4.14. Statistical Analyses

Medical records were used to retrospectively review clinical characteristics, including tumor staging, adjuvant chemotherapy, and survival outcomes. Recurrence-free survival (RFS) and overall survival (OS) were evaluated with Kaplan–Meier analysis. RFS was defined as the time from diagnosis to recurrence, metastasis, or death. OS was defined as the time from diagnosis until death from any cause. Data from patients who were alive at the time of the analysis were censored on the basis of the last recorded dates on which the patients were known to be alive. Unless noted otherwise, all experiments were performed at least in triplicate. The data are presented as the means ± standard deviations. Groups were compared via a t-test or two-way ANOVA, with *p* < 0.05 indicating statistically significant differences. All the statistical analyses were performed via GraphPad Prism version 10 (GraphPad Software Inc., La Jolla, CA, USA).

## 5. Conclusions

This study provides novel insights into how *NRXN1* influences colon cancer progression and highlights potential avenues for targeted therapy through the modulation of downstream signals, such as GSK3β inhibitors. Following our identification of *NRXN1* as a potential biomarker, further research into CNV biomarker-based diagnostic and therapeutic approaches for selecting patients at high risk of recurrence, as well as for improving patient outcomes, is warranted.

## Figures and Tables

**Figure 1 ijms-25-11423-f001:**
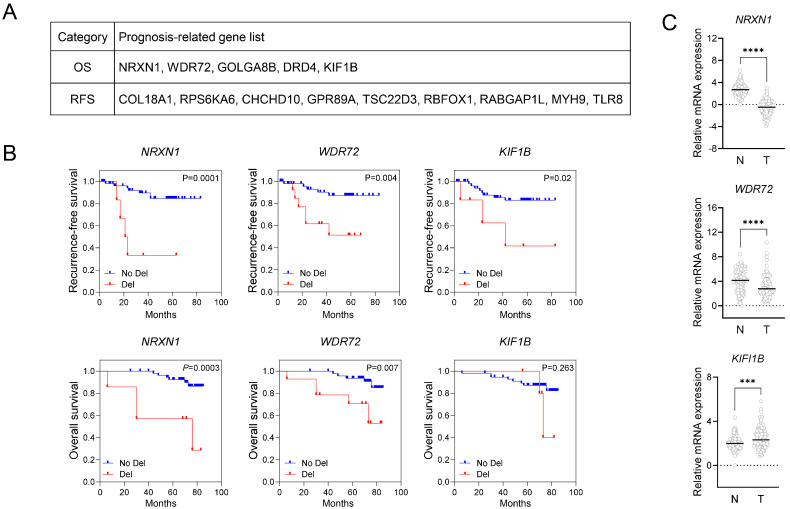
Identification of survival-related candidate genes in colorectal cancer. (**A**) Gene list of putative survival-related factors identified by ML analysis based on integrating CNV and gene expression data. (**B**) Predictive values of putative survival-related genes for CRC patients. Left: Kaplan–Meier curves for CNVs of the indicated genes associated with RFS and OC in CRC patients (stage III, *n* = 63). Kaplan–Meier plots were constructed based on the copy number status of each gene to determine the difference between cases with gene-harboring copy number deletion versus unaltered status. Blue indicates no deletion; red indicates copy number deletion. The log-rank *p*-value for significance between the curves is indicated at the top of each panel within the figure. (**C**) Differential gene expression analysis of *NRXN1*, *WDR72*, and *KIF1B* in tumor and normal tissues (stage II, III, *n* = 120). qRT-PCR determined the gene expression of the indicated genes. The expression of GAPDH was used as an internal reference control for qRT-PCR analysis. OS, overall survival; RFS, Recurrence-free survival. Data are presented as mean ± SD. *** *p* < 0.001, **** *p* < 0.0001.

**Figure 2 ijms-25-11423-f002:**
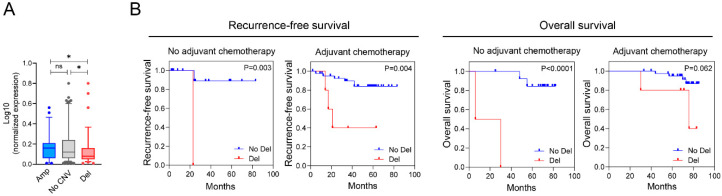
The correlation between *NRXN1* CNV and gene expression, and Kaplan–Meier survival analyses based on *NRXN1* CNV after adjuvant chemotherapy. (**A**) The correlation between CNV (in genomic DNA) and gene expression (in mRNA) of *NRXN1* in CRC (stage II, III; *n* = 120). qRT-PCR determined *NRXN1* expression. Expression of GAPDH was used as an internal reference control for RT-qPCR analysis. No CNV, no copy number variation; Amp, amplification; Del, deletion. (**B**) Kaplan–Meier plots for *NRXN1* CND associated with the recurrence-free survival and overall survival of CRC patients with or without adjuvant chemotherapy, including 5-fluorouracil and oxaliplatin. n indicates the number of cases and events. Blue indicates no CND; red indicates *NRXN1* CND. The log-rank *p*-value for significance between the curves is indicated at the bottom of each panel. *NRXN1* CND shows a significant association with poor RFS and OS in the adjuvant chemotherapy receiving group. Data are presented as mean ± SD. ns; no significant, * *p* < 0.05.

**Figure 3 ijms-25-11423-f003:**
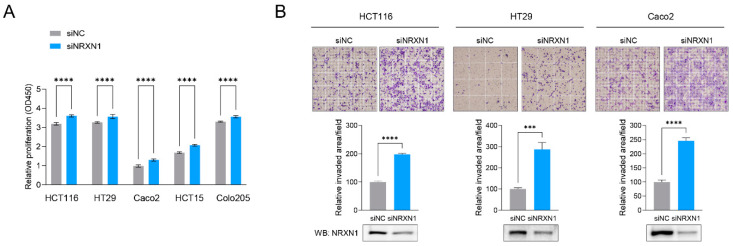
*NRXN1* knockdown enhances CRC cell viability and significantly increases invasion. Comparison of cell viability (**A**) and invasion potential (**B**) of CRC cell lines transfected with siNC or siNRXN1. Western blot analysis verifying NRXN1 protein knockdown in siNRXN1-transfected cells. Data are presented as mean ± SD. *** *p* < 0.001, **** *p* < 0.0001.

**Figure 4 ijms-25-11423-f004:**
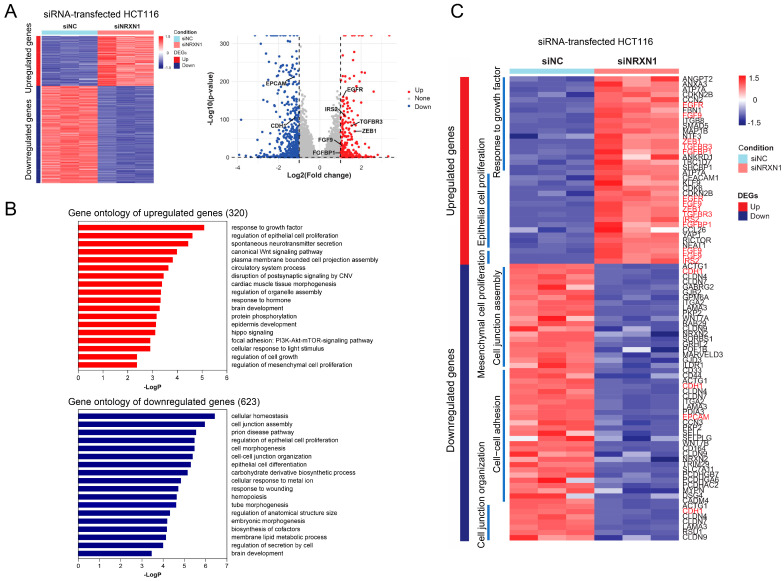
Transcriptome analysis of NRXN1 knockdown cells. (**A**) Heatmap and volcano plots showing the DEGs in siNRXN1-treated HCT116 cells compared to siNC (n = 3/group). (**B**) GO enrichment analysis of the biological process terms associated with 320 upregulated and 623 downregulated genes. (**C**) Heatmap depicting the gene expression related to the response to growth factor and regulation of epithelial cell proliferation, mesenchymal cell proliferation, cell junction assembly, cell–cell adhesion, and cell junction organization. Genes in red indicate those regulated by NRXN1 knockdown and associated with Epithelial-Mesenchymal Transition (EMT). DEG; differentially expressed genes, siNC; negative control siRNA, siNRXN1; NRXN1 siRNA.

**Figure 5 ijms-25-11423-f005:**
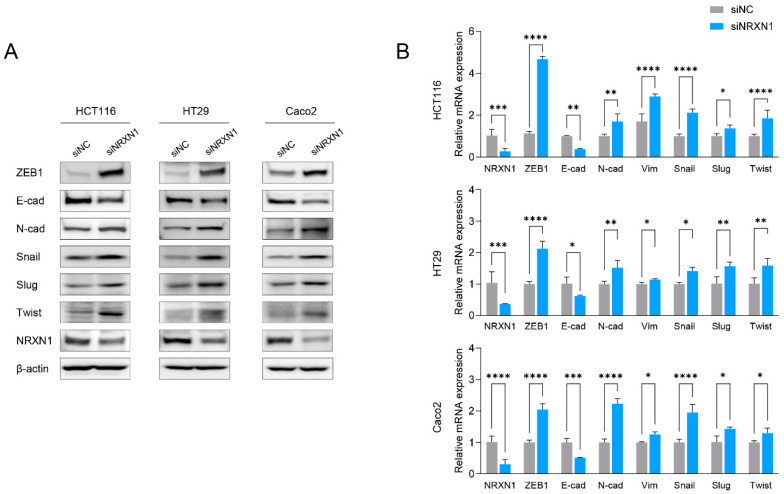
*NRXN1* knockdown induces EMT in CRC cells. Effect of siRNA-mediated *NRXN1* gene silencing on the EMT signaling pathway in CRC cells. (**A**) Gene expression of the EMT-related signaling pathway was determined using Western blot analysis with the indicated antibodies. β-actin was used as the loading control. (**B**) qRT-PCR determined the mRNA expression of EMT-related genes. siNC; negative control siRNA, siNRXN1; *NRXN1* siRNA. Data are presented as mean ± SD. * *p* < 0.05, ** *p* < 0.01, *** *p* < 0.001, **** *p* < 0.0001.

**Figure 6 ijms-25-11423-f006:**
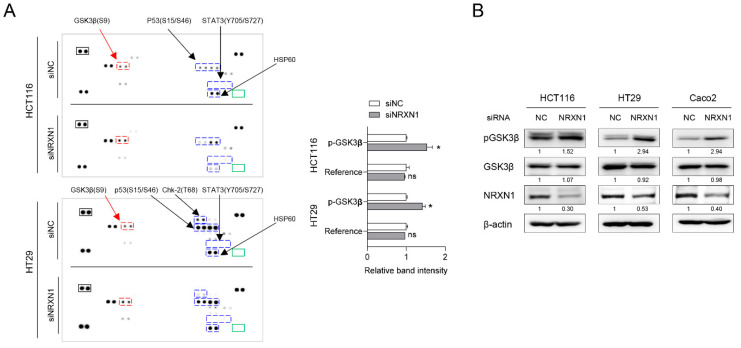
*NRXN1* knockdown induces GSK3β phosphorylation. (**A**) Left: the images of phospho-kinase arrays with marked proteins. Positive reference double spots are marked in black rectangles; the area of negative control spots is marked in green rectangles. Phosphorylated kinase spots showing the difference between siNC and siNRXN1 are marked with blue and red rectangles. Blue indicates kinases that are not commonly altered between HCT116 and HT29, while red indicates those that are consistently altered in both cell lines. Right: bar graph showing the relative change in Ser9 phospho-GSK3β and positive reference spot intensity between siNC- and siNRXN1-transfected cells. Spot intensities were quantified using ImageJ software (version 1.53k) and normalized to those of positive controls on the same membrane. Data are presented as mean ± SD. ns; not significant, * *p* < 0.05 (**B**) Western blot analysis of GSK3β phosphorylation at its residue Ser9. β-actin was used as the loading control. Band intensity quantification is labeled below the blot.

**Figure 7 ijms-25-11423-f007:**
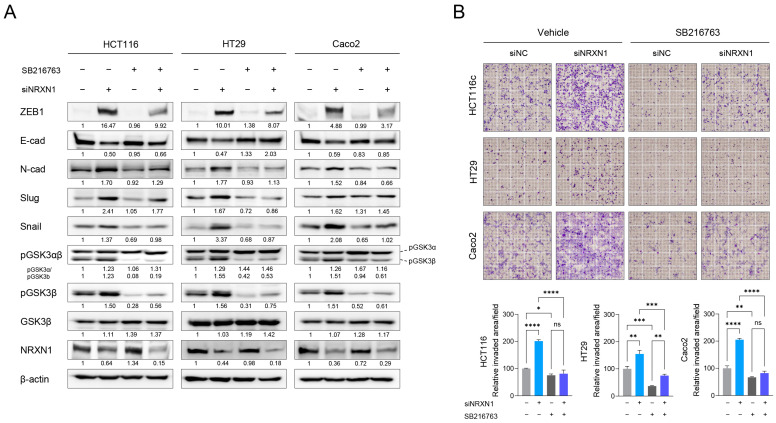
*NRXN1* knockdown promotes the invasion of CRC cell through GSK3β-mediated EMT induction. The effect of GSK3β inhibition on NRXN1 knockdown-induced EMT and invasion property in CRC. siNC or siNRXN1-transfected HCT116, HT29, and Caco2 cells were treated with GSK3β inhibitor, SB216763 (10 µM) for 24 h. (**A**) The EMT signaling pathway was evaluated by Western blot analysis with the indicated antibodies. β-actin was used as the loading control. Band intensity quantification is labeled below the blot. (**B**) Upper: representative Transwell invasion assay of siRNA-transfected CRC cells treated with vehicle or SB216763 (10 µM). Lower: quantification of invasive capacities of CRC cells. Data are presented as mean ± SD. ns; no significant, * *p* < 0.05, ** *p* < 0.01, *** *p* < 0.001, **** *p* < 0.0001.

## Data Availability

The original contributions presented in the study are included in the article/Appendix A, and further inquiries can be directed to the corresponding authors.

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
