# Peer review of "NRXN1 as a Prognostic Biomarker: Linking Copy Number Variation to EMT and Survival in Colon Cancer"

_ijms, 2024, doi:10.3390/ijms252111423_

Round 1
Reviewer 1 Report
Comments and Suggestions for Authors
The authors identified that the copy number variation of NRXN1 is significantly associated with overall survival and recurrence-free survival in colorectal cancer. NRXN1 is downregulated in tumor and knockdown upregulates cell proliferation, invasion by activation of GSK3b-mediated EMT. But there are several concerns:
1. In session 2.4, please add qPCR and WB results of NRXN1 in control and siNRXN1 cells. If knockdown of NRXN1 caused increasing of cell proliferation. How do you eliminate that effect in your cell invasion assay?
2. Figure 6 and 7, please quantify the WB results.
3. Could you add a rescue experiment showing when you add back NRXN1 in siNRXN1 cells the EMT phenotype will be reversed?
This manuscript can be accept for publish with all the concerns be addressed.
Comments on the Quality of English Language
No obvious error
Author Response
1. In session 2.4, please add qPCR and WB results of NRXN1 in control and siNRXN1 cells. If knockdown of NRXN1 caused increasing of cell proliferation. How do you eliminate that effect in your cell invasion assay?
Response: Thank you for your valuable comment. We have now included qPCR and Western Blot (WB) data to confirm the successful knockdown of NRXN1 in the siNRXN1-transfected cells compared to control cells, as shown in the revised manuscript (Figure 3B and supplementary Figure 2).
Regarding the concern about NRXN1 knockdown potentially increasing cell proliferation and its impact on the invasion assay, we took several steps to address this:
- Seeding equal numbers of cells: We ensured that an equal number (1 × 10⁵) of control and siNRXN1-transfected cells were counted and seeded for the invasion assay to eliminate any discrepancies caused by differences in cell number
- Low-serum condition: The invasion assay was conducted under low serum conditions (0.5% bovine serum albumin) to minimized the potential influence of proliferation during the invasion assay. This detail has mentioned in the revised manuscript (line 351-354).
Furthermore, the RNA sequencing and WB data presented in Figures 4 and 5 provide strong evidence that NRXN1 knockdown increases the invasion potential of colorectal cancer cells, independent of proliferation effects.
2. Figure 6 and 7, please quantify the WB results.
Response: Thank you for your valuable comment. We have quantified the Western blot (WB) results for Figures 6 and 7. The quantification of band intensity is now included in the corresponding Western blot figures. The quantification results are provided as relative values normalized to β-actin, as the loading control. We appreciate your comments, which has contributed to improving the clarity and thoroughness of our findings.
3. Could you add a rescue experiment showing when you add back NRXN1 in siNRXN1 cells the EMT phenotype will be reversed?
Response:
We sincerely appreciate your insightful comment regarding the rescue experiment to assess the effects of restoring NRXN1 expression on the EMT phenotype induced by NRXN1 knockdown. Based on your valuable suggestion, we initially aimed to conduct this experiment and requested an extension to perform the additional work required. However, we have yet to receive a response to our extension request, and due to the limited revision timeframe, we were unable to conduct the rescue experiment prior to the submission of our revised manuscript.
We recognize the significance of this experiment in further clarifying NRXN1’s role in EMT and intend to explore this in future studies. Investigating the effects of restoring NRXN1 expression is an important direction for our research, and we plan to incorporate this into upcoming experiments. Your constructive feedback continues to shape the progress of our research, and we greatly appreciate your valuable suggestion.

Reviewer 2 Report
Comments and Suggestions for Authors
Major comments:
- CNVs have been previously linked to EMT-related biomarkers in colon cancers (for e.g. https://www.ncbi.nlm.nih.gov/pmc/articles/PMC9013447/).Assuch the paper lacks much needed novelty and mechanistic depth.
- The mention in the abstract that – “Recently, by using machine learning to analyze CNV and gene expression data, we identified potential survival-related genes for colorectal cancer (CRC).” This was from a previous publication. Please use the methods section to inform that these targets were already identified from a previous publication as it comes across that machine learning approach was utilized in this manuscript, which is not the case here.
- For results section 2.3, the authors state that – “This analysis revealed that in the NRXN1 CND group, patients who received adjuvant chemotherapy, such as 5-fluorouracil and oxaliplatin, had poor RFS and OS (Figure 2B), indicating that the NRXN1 CND may be involved in chemotherapy resistance and tumor recurrence.” The conclusion should also be made about no adjuvant chemotherapy arm with or without NRXN1 CND.
- The authors state that – “Compared with the siNC-transfected control cells, the siNRXN1-transfected CRC cell lines presented a greater proliferation rate (Figure 3A)”. Figure 3A suggests that the increase in “proliferation” is modest, to say the least. Additionally, the CCK8 assay measures cell viability (mitochondrial activity). Using a gentle voice would be nice. Moreover, a more rigorous method of CFSE staining should be used to measure the difference in cell proliferation.
- Whole blot images should be provided for all the WB analyses as separate files as per the journal guidelines for Figures 5, 6, and 7.
- The authors make a case for GSK3 beta to mediate the EMT and invasion upon NRXN1 KD in 3 cell lines by using inhibitor SB216763 at 10 micromolar concentration. To add depth and more clarity to experiments, selective siRNA KD or genetic overexpression experiments of GSK3 beta should be done. At this concentration, other kinase can be inhibited as well.
- There should be a model to explain the findings. If GSk3 beta is mediating these effects, by doing ChIP assays one can determine how EMT factors are being influenced.
- There is a need to show the in vivo significance of these results in CRC mouse models.
Minor comments:
- Figures 2 and 4 are too small and need to be made more legible. For fig2, include the X-axis. The authors should use either relapse-free survival or recurrence-free survival on Y-axis.
- Provide a reference for this sentence on line 165 – “ZEB1 induces EMT during cancer metastasis by transcriptionally repressing CDH1, which encodes E-cadherin.”
Comments on the Quality of English LanguageFine
Author Response
Major comments:
1. CNVs have been previously linked to EMT-related biomarkers in colon cancers (for e.g. https://www.ncbi.nlm.nih.gov/pmc/articles/PMC9013447/). As such the paper lacks much needed novelty and mechanistic depth.
Response: We sincerely appreciate your insightful comment regarding the connection between CNVs and EMT-related biomarkers in colon cancers. We recognize the significance of the referenced study (PMID: 9013447), which utilizes tools like GEPIA2 and TIMER to analyze public data and identify key CNV-related biomarkers, such as CDKN2A, CMTM8, and ILK. We have added this important work as a reference in our revised manuscript and fully acknowledge its significant contributions to the field (line 222).
Our study offers several novel contributions, which we would like to emphasize:
First, we identified survival-related gene candidates through machine learning in our previous study and directly validated these findings using CNV analysis on patient samples in the current study. While machine learning allowed us to identify potential survival-associated genes, experimental validation was essential to confirm their value as biomarkers. By conducting expression and CNV analysis on patient samples, NRXN1 was identified as the most promising biomarker, showing a significant association with both overall survival (OS) and recurrence-free survival (RFS).
Second, we conducted biological function studies using colorectal cancer cell lines to explore NRXN1’s role, adding significant novelty to our research. In addition to the functional analysis, we performed mechanistic studies, which demonstrated that NRXN1 knockdown enhances colorectal cancer invasion. Transcriptome analysis and in vitro experiment revealed that these effects are mediated through GSK3β-dependent epithelial–mesenchymal transition (EMT), providing critical mechanistic insights into NRXN1’s involvement in colorectal cancer progression.
By identifying and functionally characterizing NRXN1, our study adds to the growing diversity of CNV-associated biomarkers, providing new opportunities for predicting recurrence and enhancing treatment outcomes in colorectal cancer.
These findings underscore the novelty and clinical relevance of NRXN1 as a biomarker, offering new opportunities for predicting recurrence and improving treatment outcomes in colorectal cancer.
2. The mention in the abstract that – “Recently, by using machine learning to analyze CNV and gene expression data, we identified potential survival-related genes for colorectal cancer (CRC).” This was from a previous publication. Please use the methods section to inform that these targets were already identified from a previous publication as it comes across that machine learning approach was utilized in this manuscript, which is not the case here.
Response: Thank you for your insightful comment. We have revised the manuscript to clarify in the methods section of abstract that the survival-related genes were identified in our previous publication using a machine learning approach (lines 45-46).
3. For results section 2.3, the authors state that – “This analysis revealed that in the NRXN1 CND group, patients who received adjuvant chemotherapy, such as 5-fluorouracil and oxaliplatin, had poor RFS and OS (Figure 2B), indicating that the NRXN1 CND may be involved in chemotherapy resistance and tumor recurrence.” The conclusion should also be made about no adjuvant chemotherapy arm with or without NRXN1 CND.
Response: Thank you for your valuable comment. We have revised section 2.3 to clarify that the NRXN1 CND group showed poor RFS and OS compared to the no deletion group, regardless of whether patients received chemotherapy. This revision is reflected in the updated manuscript (lines 133-137).
4. The authors state that – “Compared with the siNC-transfected control cells, the siNRXN1-transfected CRC cell lines presented a greater proliferation rate (Figure 3A)”. Figure 3A suggests that the increase in “proliferation” is modest, to say the least. Additionally, the CCK8 assay measures cell viability (mitochondrial activity). Using a gentle voice would be nice. Moreover, a more rigorous method of CFSE staining should be used to measure the difference in cell proliferation.
Response: Thank you for your thoughtful comment and suggestion. In response, we have revised the wording to be more neutral tone (line 144).
NRXN1 knockdown resulted in a statistically significant increase in proliferation, as shown in Figure 3A. However, our RNA-seq and western blot results revealed more pronounced changes in invasion property and EMT. Based on these findings, we prioritized investigating the EMT-related mechanisms induced by NRXN1 knockdown rather than performing CFSE staining, which is a more rigorous method for measuring proliferation.
5. Whole blot images should be provided for all the WB analyses as separate files as per the journal guidelines for Figures 5, 6, and 7.
Response: Thank you for your helpful comment. Following the IJMS journal guidelines, we have provided the uncropped original images of the western blots for Figures 3, 5, 6, 7, and S2. These images have been organized and uploaded separately by figure, with size markers. The files have been uploaded as supplementary materials.
6. The authors make a case for GSK3 beta to mediate the EMT and invasion upon NRXN1 KD in 3 cell lines by using inhibitor SB216763 at 10 micromolar concentration. To add depth and more clarity to experiments, selective siRNA KD or genetic overexpression experiments of GSK3 beta should be done. At this concentration, other kinase can be inhibited as well.
Response: We appreciate the reviewer’s suggestion to explore selective siRNA knockdown or genetic overexpression experiments of GSK3β. SB216763 is a potent inhibitor of both GSK3α and GSK3β, and we acknowledge the potential for its non-specific effects. To address this, we examined the phosphorylation levels of both GSK3α and GSK3β (pGSK3α/β) under our experimental conditions. Our data confirmed that SB216763 selectively inhibits GSK3β phosphorylation without affecting GSK3α. We have updated Figure 7 to reflect unchanged pGSK3α levels, confirming SB216763’s specificity.
Our aim was to inhibit the phosphorylation of GSK3β following NRXN1 knockdown, not to reduce total GSK3β protein levels. siRNA-mediated knockdown of GSK3β would reduce its overall protein level, which could introduce additional effects unrelated to phosphorylation. Therefore, the SB216763 treatment results provide more precise insights into the role of phosphorylated GSK3β in promoting EMT.
7. There should be a model to explain the findings. If GSk3 beta is mediating these effects, by doing ChIP assays one can determine how EMT factors are being influenced.
Response: We sincerely appreciate the reviewer's insightful suggestion to explore how GSK3β influences EMT factors through ChIP assays. While we acknowledge the importance of such assays in investigating transcriptional regulation, our study, supported by functional experiments and relevant literature, provides substantial evidence of GSK3β’s involvement in mediating EMT.
Our data show that NRXN1 knockdown increases phosphorylation of GSK3β at Ser9, which promotes mesenchymal marker expression (Snail, Slug, and N-cadherin) and reduces epithelial markers like E-cadherin. This is consistent with the well-established role of GSK3β phosphorylation in inactivating its suppression of EMT transcription factors, particularly Snail, as described by Zhou et al. (2004) and Vijay et al. (2019).
Furthermore, we observed that treatment with a GSK3β inhibitor reversed EMT, further confirming the critical role of GSK3β phosphorylation in this process. Additionally, Lee et al. (2012) demonstrated that the PKC/GSK3β pathway plays a key role in EGF-induced EMT, underscoring the broader significance of GSK3β phosphorylation in cancer cell EMT progression.
We have carefully incorporated these references and additional explanations (lines 241-246) into our manuscript revision to offer a more comprehensive explanation of the underlying mechanisms. We hope this revised discussion sufficiently addresses the reviewer's thoughtful comments without the need for additional ChIP assays.
References
- Zhou BP, Deng J, Xia W, et al. "Dual regulation of Snail by GSK-3beta-mediated phosphorylation in control of epithelial-mesenchymal transition." Nat Cell Biol. 2004;6(10):931-40.
- Vijay GV, Zhao N, Den Hollander P, et al. "GSK3β regulates epithelial-mesenchymal transition and cancer stem cell properties in triple-negative breast cancer." Breast Cancer Res. 2019;21(1):37.
- Zong-cai Liu, Xiao-hui Chen, et al. "Snail regulated by PKC/GSK-3β pathway is crucial for EGF-induced epithelial-mesenchymal transition (EMT) of cancer cells." Cell Tissue Res. 2014; 358(2):491-502
8. There is a need to show the in vivo significance of these results in CRC mouse models.
Response: We sincerely appreciate the reviewer's recommendation to assess the in vivo significance of our findings using CRC mouse models. While we agree that such experiments could provide valuable insights, the time constraints make it challenging to conduct them at this stage. To perform in vivo studies, we first must generate NRXN1 stably knockdown cell lines using shRNA-NRXN1 constructs. This process, including shRNA design, construction production, and establishment of stable cell lines, typically takes at least three months.
Given this timeline, in vivo experiments cannot be completed within the current scope of our study. However, we recognize the importance of these experiments and are committed to pursuing them in our future research.
Minor comments:
1. Figures 2 and 4 are too small and need to be made more legible. For fig2, include the X-axis. The authors should use either relapse-free survival or recurrence-free survival on Y-axis.
Response: We greatly appreciate the reviewer's feedback regarding Figures 2 and 4. In response, we have increased the size of both figures and improved their legibility by enhancing the resolution. For Figure 2, we have added the X-axis, as suggested, to ensure a more precise data presentation. We have also standardized the Y-axis labeling by consistently using "recurrence-free survival" throughout the manuscript. Thanks to the reviewer’s valuable suggestion, we were able to make these improvements to enhance clarity.
2. Provide a reference for this sentence on line 165 – “ZEB1 induces EMT during cancer metastasis by transcriptionally repressing CDH1, which encodes E-cadherin.”
Response: Thank you for your detailed comment. We have revised the manuscript and added a reference to support the sentence (lines 160-162) We appreciate the reviewer’s suggestion, which allowed us to strengthen the accuracy of this section.

Round 2
Reviewer 2 Report
Comments and Suggestions for Authors
2. The mention in the abstract that – “Recently, by using machine learning to analyze CNV and gene expression data, we identified potential survival-related genes for colorectal cancer (CRC).” This was from a previous publication. Please use the methods section to inform that these targets were already identified from a previous publication as it comes across that machine learning approach was utilized in this manuscript, which is not the case here.
Response: Thank you for your insightful comment. We have revised the manuscript to clarify in the methods section of abstract that the survival-related genes were identified in our previous publication using a machine learning approach (lines 45-46).
Query:
There is no mention of such comments in lines 45-46.
3. For results section 2.3, the authors state that – “This analysis revealed that in the NRXN1 CND group, patients who received adjuvant chemotherapy, such as 5-fluorouracil and oxaliplatin, had poor RFS and OS (Figure 2B), indicating that the NRXN1 CND may be involved in chemotherapy resistance and tumor recurrence.” The conclusion should also be made about no adjuvant chemotherapy arm with or without NRXN1 CND.
Response: Thank you for your valuable comment. We have revised section 2.3 to clarify that the NRXN1 CND group showed poor RFS and OS compared to the no deletion group, regardless of whether patients received chemotherapy. This revision is reflected in the updated manuscript (lines 133-137).
Query: From Fig 2A (left panel), for the No Adjuvant Chemotherapy category, it seems that there is not much difference in the RFP or OS as far as NRXN1 CND is concerned. How do the authors reconcile this?
4. The authors state that – “Compared with the siNC-transfected control cells, the siNRXN1-transfected CRC cell lines presented a greater proliferation rate (Figure 3A)”. Figure 3A suggests that the increase in “proliferation” is modest, to say the least. Additionally, the CCK8 assay measures cell viability (mitochondrial activity). Using a gentle voice would be nice. Moreover, a more rigorous method of CFSE staining should be used to measure the difference in cell proliferation.
Response: Thank you for your thoughtful comment and suggestion. In response, we have revised the wording to be more neutral tone (line 144).
NRXN1 knockdown resulted in a statistically significant increase in proliferation, as shown in Figure 3A. However, our RNA-seq and western blot results revealed more pronounced changes in invasion property and EMT. Based on these findings, we prioritized investigating the EMT-related mechanisms
induced by NRXN1 knockdown rather than performing CFSE staining, which is a more rigorous method for measuring proliferation.
Query: There is no statement mentioned on line 144 regarding this. Line 148 states that – “Compared with the siNC-transfected control 147 cells, the siNRXN1-transfected CRC cell lines presented increased proliferation rate (Fig- 148 ure 3A).” As mentioned earlier, a CFSE staining has to be done to state that proliferation is affected.
5. Whole blot images should be provided for all the WB analyses as separate files as per the journal guidelines for Figures 5, 6, and 7.
Response: Thank you for your helpful comment. Following the IJMS journal guidelines, we have provided the uncropped original images of the western blots for Figures 3, 5, 6, 7, and S2. These images have been organized and uploaded separately by figure, with size markers. The files have been uploaded as supplementary materials.
Query: No uncropped original images of the western blots for Figures 3, 5, 6, and 7 have been provided!
6. The authors make a case for GSK3 beta to mediate the EMT and invasion upon NRXN1 KD in 3 cell lines by using inhibitor SB216763 at 10 micromolar concentration. To add depth and more clarity to experiments, selective siRNA KD or genetic overexpression experiments of GSK3 beta should be done. At this concentration, other kinase can be inhibited as well.
Response: We appreciate the reviewer’s suggestion to explore selective siRNA knockdown or genetic overexpression experiments of GSK3β. SB216763 is a potent inhibitor of both GSK3α and GSK3β, and we acknowledge the potential for its non-specific effects. To address this, we examined the phosphorylation levels of both GSK3α and GSK3β (pGSK3α/β) under our experimental conditions. Our data confirmed that SB216763 selectively inhibits GSK3β phosphorylation without affecting GSK3α. We have updated Figure 7 to reflect unchanged pGSK3α levels, confirming SB216763’s specificity.
Our aim was to inhibit the phosphorylation of GSK3β following NRXN1 knockdown, not to reduce total GSK3β protein levels. siRNA-mediated knockdown of GSK3β would reduce its overall protein level, which could introduce additional effects unrelated to phosphorylation. Therefore, the SB216763 treatment results provide more precise insights into the role of phosphorylated GSK3β in promoting EMT.
Query: While the blots in Fig 7 support the author's comments nicely, it is still worth noting that other protein kinases can be affected at this concentration. This is why using a siRNA KD or genetic overexpression experiments of GSK3 beta would be a cleaner approach.
Author Response
OCT 15, 2024
Rebuttal letter
Manuscript ID: ijms-3196883
Title: "NRXN1 as a Prognostic Biomarker: Linking Copy Number Variation to EMT"
Authors: Hyun-Jin Bang*, Hyun-Jeong Shim1*, Mi-Ra Park1, Sumin Yoon3, Kyung Hyun Yoo3, Young-Kook Kim4, Hyunju Lee5, Jeong-Seok Nam6, Jun-Eul Hwang1, Woo-Kyun Bae1, 2, Ik-Joo Chung1,2, Eun-Gene Sun1,2** and Sang-Hee Cho1**
Dear Editor and Reviewers,
We are pleased to submit the revised version of our manuscript titled “NRXN1 as a Prognostic Biomarker: Linking Copy Number Variation to EMT" (ijms-3196883) for consideration in International Journal of Molecular Sciences.
We thank you and the reviewers for your time and thoughtful feedback on our manuscript. We sincerely appreciate the constructive comments and suggestions, which have significantly enhanced our work's clarity and scientific rigor. We have carefully addressed all the points raised and made the necessary revisions. These improvements have strengthened our study, and we hope it is suitable for publication. Thank you again for your consideration, and we look forward to your favorable response.
Thank you again for your consideration, and we look forward to your favorable response.
Sincerely,
Sang-Hee Cho
Chonnam National University Medical School and Hwasun Hospital, Hwasun,
shcho@chonnam.ac.kr
RESPONSE TO REVIEWER COMMENTS
2. The mention in the abstract that – “Recently, by using machine learning to analyze CNV and gene expression data, we identified potential survival-related genes for colorectal cancer (CRC).” This was from a previous publication. Please use the methods section to inform that these targets were already identified from a previous publication as it comes across that machine learning approach was utilized in this manuscript, which is not the case here.
Response: Thank you for your insightful comment. We have revised the manuscript to clarify in the methods section of abstract that the survival-related genes were identified in our previous publication using a machine learning approach (lines 45-46).
Query: There is no mention of such comments in lines 45-46.
Response 2: Thank you for your detailed comment. We have revised the manuscript to clearly state in the Methods section of the abstract that the survival-related genes were identified using a machine-learning approach in our previous study (lines 45-46). Additionally, we have added a reference to our previous study in the Methods section (lines 280-281) to clarify further. We apologize for the confusion and appreciate your thorough review.
3. For results section 2.3, the authors state that – “This analysis revealed that in the NRXN1 CND group, patients who received adjuvant chemotherapy, such as 5-fluorouracil and oxaliplatin, had poor RFS and OS (Figure 2B), indicating that the NRXN1 CND may be involved in chemotherapy resistance and tumor recurrence.” The conclusion should also be made about no adjuvant chemotherapy arm with or without NRXN1 CND.
Response: Thank you for your valuable comment. We have revised section 2.3 to clarify that the NRXN1 CND group showed poor RFS and OS compared to the no deletion group, regardless of whether patients received chemotherapy. This revision is reflected in the updated manuscript (lines 133-137).
Query: From Fig 2A (left panel), for the No Adjuvant Chemotherapy category, it seems that there is not much difference in the RFP or OS as far as NRXN1 CND is concerned. How do the authors reconcile this?
Response 2: Thank you for your thoughtful comment. As you observed, in the NRXN1 CND group, patients who did not receive adjuvant chemotherapy had worse RFS (p=0.003) and OS (p<0.0001). Similarly, patients who received adjuvant chemotherapy also showed poor outcomes, with no significant survival benefit or prevention of recurrence compared to those who did not receive chemotherapy. This suggests that adjuvant chemotherapy had a negligible impact on improving survival in the NRXN1 CND group. We have revised the manuscript and Figure 2B to clarify these findings; updates are reflected in the manuscript (lines 132-141).
4. The authors state that – “Compared with the siNC-transfected control cells, the siNRXN1-transfected CRC cell lines presented a greater proliferation rate (Figure 3A)”. Figure 3A suggests that the increase in “proliferation” is modest, to say the least. Additionally, the CCK8 assay measures cell viability (mitochondrial activity). Using a gentle voice would be nice. Moreover, a more rigorous method of CFSE staining should be used to measure the difference in cell proliferation.
Response: Thank you for your thoughtful comment and suggestion. In response, we have revised the wording to be more neutral tone (line 144). NRXN1 knockdown resulted in a statistically significant increase in proliferation, as shown in Figure 3A. However, our RNA-seq and western blot results revealed more pronounced changes in invasion property and EMT. Based on these findings, we prioritized investigating the EMT-related mechanisms induced by NRXN1 knockdown rather than performing CFSE staining, which is a more rigorous method for measuring proliferation.
Query: There is no statement mentioned on line 144 regarding this. Line 148 states that – “Compared with the siNC-transfected control 147 cells, the siNRXN1-transfected CRC cell lines presented increased proliferation rate (Fig- 148 ure 3A).” As mentioned earlier, a CFSE staining has to be done to state that proliferation is affected.
Response 2: Thank you for your valuable suggestion regarding using a rigorous method to assess proliferation. In response, we performed CFSE analysis, as recommended, to evaluate proliferation in three different cell lines. However, the CFSE results showed no significant difference in proliferation between the control (siNC-transfected) and NRXN1 knockdown groups across all three cell lines (Figure R1). Based on these findings, we have revised the manuscript to replace “proliferation” with “viability,” as the CCK8 assay primarily measures cell viability. Furthermore, we revised the results to emphasize that NRXN1 knockdown has a pronounced effect on invasion, based on its slight effect on viability. We sincerely appreciate your insightful feedback, which has allowed us to improve the accuracy and clarity of the manuscript. The revised version has reflected the necessary changes (lines 53, 143, 148-153, 340-341, and the Figure 3 legend). We hope these revisions address your concern.
Figure R1. CSFE staining analysis of cell proliferation in siNC- and siNRXN1-transfected CRC cell lines. CFSE-labeled cells were analyzed by flow cytometry to assess the proliferation rate. The results indicate no significant difference in CFSE fluorescence intensity between the siNC control group and the siNRXN1 knockdown group across three CRC cell lines, suggesting that NRXN1 knockdown does not affect the proliferation rate.
5. Whole blot images should be provided for all the WB analyses as separate files as per the journal guidelines for Figures 5, 6, and 7.
Response: Thank you for your helpful comment. Following the IJMS journal guidelines, we have provided the uncropped original images of the western blots for Figures 3, 5, 6, 7, and S2. These images have been organized and uploaded separately by figure, with size markers. The files have been uploaded as supplementary materials.
Query: No uncropped original images of the western blots for Figures 3, 5, 6, and 7 have been provided!
Response 2: We have previously uploaded the images for Figures 3, 5, 6, 7, and S2 following the journal guidelines. However, to ensure no issues with the submission, we have uploaded the uncropped original images again, including the image for the newly added supplementary figure S3. These images have been organized and uploaded separately by figure, with size markers.
6. The authors make a case for GSK3 beta to mediate the EMT and invasion upon NRXN1 KD in 3 cell lines by using inhibitor SB216763 at 10 micromolar concentration. To add depth and more clarity to experiments, selective siRNA KD or genetic overexpression experiments of GSK3 beta should be done. At this concentration, other kinase can be inhibited as well.
Response: We appreciate the reviewer’s suggestion to explore selective siRNA knockdown or genetic overexpression experiments of GSK3β. SB216763 is a potent inhibitor of both GSK3α and GSK3β, and we acknowledge the potential for its non-specific effects. To address this, we examined the phosphorylation levels of both GSK3α and GSK3β (pGSK3α/β) under our experimental conditions. Our data confirmed that SB216763 selectively inhibits GSK3β phosphorylation without affecting GSK3α. We have updated Figure 7 to reflect unchanged pGSK3α levels, confirming SB216763’s specificity.
Our aim was to inhibit the phosphorylation of GSK3β following NRXN1 knockdown, not to reduce total GSK3β protein levels. siRNA-mediated knockdown of GSK3β would reduce its overall protein level, which could introduce additional effects unrelated to phosphorylation. Therefore, the SB216763 treatment results provide more precise insights into the role of phosphorylated GSK3β in promoting EMT.
Query: While the blots in Fig 7 support the author's comments nicely, it is still worth noting that other protein kinases can be affected at this concentration. This is why using a siRNA KD or genetic overexpression experiments of GSK3 beta would be a cleaner approach.
Response 2: We are profoundly grateful for your insightful comment, which has significantly impacted our research. Following your recommendation, we conducted additional experiments using siRNA to specifically knock down GSK3β to address the concern that SB216763 treatment could affect other protein kinases. As shown in Supplementary Figure 3, our results demonstrated that NRXN1 knockdown induced both invasion and EMT. Moreover, consistent with the effects of the GSK3β inhibitor, SB216763, siRNA-mediated knockdown of GSK3β effectively suppressed the invasion properties and EMT induced by NRXN1 knockdown. We also verified the successful knockdown of GSK3β through RT-qPCR and western blot analysis following siGSK3β transfection. These findings have been included in the revised manuscript (lines 198-203).
Additionally, we have included the siRNA and primer information related to the siGSK3β experiment in the Method section (lines 334-335) and supplementary information. This experiment, performed based on your valuable suggestion, has significantly strengthened our findings and provided more explicit evidence that GSK3β mediates NRXN1 knockdown-induced invasion and EMT. We sincerely appreciate your suggestion, which has dramatically reinforced our conclusions.
